# The Effect of Computerized Testing on Sun Bear Behavior and Enrichment Preferences

**DOI:** 10.3390/bs6040019

**Published:** 2016-09-22

**Authors:** Bonnie M. Perdue

**Affiliations:** Department of Psychology, Agnes Scott College, 141 E. College Avenue, Decatur, GA 30030, USA; Bperdue@agnesscott.edu; Tel.: +1-404-471-6268

**Keywords:** sun bear, comparative cognition, touchscreen computer, stereotypy

## Abstract

The field of comparative cognition investigates species’ differences and similarities in cognitive abilities, and sheds light on the evolutionary origins of such capacities. Cognitive testing has been carried out in a variety of species; however, there are some taxa that are underrepresented in this field. The current work follows on a recent increase in cognitive research in the order Carnivora with a specific focus on sun bears. Sun bears are the smallest existing bear species and live in tropical regions of Southeast Asia. They have an omnivorous diet and use their tongues to forage for insects and sap. Little is known about sun bear cognition, although much like other bear species, anecdotes suggest a high level of intelligence. The current work explored training sun bears to use a touchscreen computer. This effort allows for insight into cognitive abilities as well as providing a complex source of enrichment for the bears. The bears use their tongues to respond to a touchscreen computer, and the effects on stereotypic behaviors on exhibit and preference for this over other forms of enrichment were examined. Overall, bears performed well on the task and showed a preference for the computer.

## 1. Introduction

Cognitive research in nonhuman animals has a long history and has provided important insights into the evolution of cognitive abilities [1]. The field of comparative cognition involves testing for various cognitive abilities such as memory, perception, learning, and problem solving in a variety of species. Although many species have been tested for abilities such as these, a large portion of the existing literature has focused on a smaller subset of animals including nonhuman primates, rats, and pigeons. Some genera have been largely excluded from cognitive research—most likely as a result of limited access to these animals. For example, there is relatively little known about bear cognition. This has begun to change in recent years as more empirical studies have been carried out with giant pandas [2,3,4,5,6] and black bears [7,8,9,10,11,12,13,14]. 

The expansion of cognitive research in another diverse group such as bears will allow for interesting and important tests of evolutionary hypotheses that have been developed with other orders. Socioecology focuses on how various social and environmental factors might influence a variety of behaviors and cognitive abilities. For example, it has been predicted that a species’ diet may underlie variability in cognitive ability [15]. This has been tested within primates, but a fuller test of the theory would arise from examining this relationship in another taxa such as bears. Among bears, there is variability ranging from the almost entirely herbivorous giant panda to the almost entirely carnivorous polar bear, and this variability presents an interesting opportunity to expand socioecological theory. Another example relates to how mating systems might influence sex differences in spatial cognition. The range size hypothesis predicts that species with a mating system that selects for a differential range size, such as a polygynous mating system, will show enhanced sex differences in spatial cognition compared to monogamous species in which the range size is equivalent [16,17,18]. Recent work with giant pandas and Asian small-clawed river otters supported the predictions within the order Carnivora [5]. Further research comparing different bear species should allow for additional insights into the origins of various cognitive abilities.

Given the importance of expanding cognitive research to bears, it is imperative that researchers continue to develop methods to carry out systematic research. However, such testing will rely on the development and use of methods that can be widely applied to different species in such a manner that allows for careful control over experimental parameters. One such approach involves the use of computers. Computerized testing has a long history in laboratory settings, and recent work has begun to expand this to a zoo setting and allow for direct comparisons to laboratory findings [19]. Vonk and colleagues have greatly expanded the use of computerized testing with bears [7,8,9,10,11,12,13,14]. Here, this is expanded to a previously untested species—the sun bear. 

The sun bear is a small bear species found in Southeast Asia. Wild populations are often threatened by human activity including harvesting for medicinal purposes. One priority for enhancing conservation efforts is to improve education and knowledge about the animals [20], and zoos aim to educate the public about various species and conservation needs. However, captive wildlife in zoos are carefully monitored to minimize any issues that harm animal welfare. Many carnivore species, including bears, often show stereotypic behaviors such as pacing in captivity. Research is critical to identifying social or environmental approaches that aid such an effort to minimize suboptimal behaviors. This presents an interesting intersection of cognitive research and animal welfare. Recent work has suggested that cognitive testing might be beneficial for animals in captivity [21]. A persistent challenge for more traditional environmental enrichment [22] is that subjects quickly habituate and the enrichment does not offer novelty. Computers are perfectly suited to provide novel and complex challenges that are more resistant to habituation than traditional enrichment devices [23,24,25]. If sun bears can be trained to use a touchscreen computer, this would provide insights into the evolution of cognitive abilities as well as a potentially valuable form of enrichment for the bears. The present study focused on the appropriateness of this technique in this species as well as an assessment of its potentially enriching influence via behavioral observations and a preference assessment. 

## 2. Method

### 2.1. Subjects

Testing took place with two sun bears, Sabah (female, 18 YO) and Xander (male, 19 YO), housed at Zoo Atlanta in Atlanta, Georgia. They were housed according to zoo protocol and provided with ad libitum access to water and their regular diet throughout the testing days. These two subjects have been housed at Zoo Atlanta since 2010 in the Trader’s Alley exhibit. They are socially housed, but separated for testing. Participation in testing was voluntary and relied on subjects shifting into the test enclosure (one of their indoor enclosures). No formal cognitive tasks have been presented to these subjects, but they do engage in regular training and enrichment protocols.

### 2.2. Procedure

#### 2.2.1. Computer Training

The apparatus consisted of an aluminum shell that housed the computer, a pellet dispenser, a power source, and a touchscreen monitor. This was mounted on the enclosure in such a way that the only non-aluminum surface the bears could access was the touchscreen itself. An infrared screen from Touchmonitor (CarrollTouch Touch Technology, Dual Serial/USB Touch Interface) was used. Bears could activate the screen by breaking the plane with their tongues (see below for details on training). A correct response yielded a flavored pellet from BioServ (flavors included chocolate, marshmallow, pina colada, very berry, and banana). Bears were tested approximately once per week from February 2015 to October 2015. The animal carestaff shifted animals into the enclosure and participation in testing was voluntary. 

For the first trials, dots of honey were spread on the screen to encourage interaction with the touchscreen. Any touch to the screen yielded the delivery of a pellet (in addition to the honey itself). Initially, the entire screen was red, and any touch would yield a tone and pellet delivery. Once subjects were reliably using their tongues to touch the screen, the program was then updated so that only half of the screen was red (either split on the vertical or horizontal dimension and randomly selected for each trial). Once proficiency with this stage was achieved, the red space became even smaller until a very small square was touched with the tongue. Any touches to non-red areas of the screen yielded a different tone, and no food was delivered. Thus, subjects were gradually shaped to respond to a particular stimulus on the screen. After this was achieved, the program allowed for movement to be initiated, and the red square would move randomly on the screen. Again, subjects had to touch the red square to receive reinforcement.

After this initial training to use the computer, subjects were presented with a standard two-choice discrimination task using clip-art stimuli. In this task, two stimuli appeared in random locations on the screen, with one being selected as the correct response. Touches to the S+ yielded a pellet and a tone, whereas touches to the S- yielded a brief timeout and a different tone. 

#### 2.2.2. Behavioral Observations

Subjects were observed on exhibit (testing took place off-exhibit) for a period prior to the introduction of the computer (June 2014–January 2015) and once computer testing had begun (February 2015–October 2015). Focal-animal data were collected using instantaneous and all-occurrence sampling between 1000–1200 and 1400–1600. A variety of behaviors were observed including aggressive behaviors (both contact and non-contact aggression) and stereotypic behaviors (pacing and backwards walking, an abnormal locomotive pattern) (see Appendix A for a detailed ethogram). Observers were undergraduate students trained for behavioral observations until their coding of behavior reached 85% agreement with the author. The variables of external temperature, time of day, and number of enrichment items present in the enclosure were also recorded during each session. Data for the “pre-computer” phase were collected from June 2014 to January 2015, and data for the “computer phase” were collected from February 2015 to October 2015. Comparisons of various behaviors before and after the introduction of the computer were made using independent sample *t*-tests conducted separately for each bear. 

#### 2.2.3. Preference Assessment

As a post-hoc analysis of preference, each bear was presented with a total of 4 trials in which the computer was available in the enclosure in addition to a standard enrichment device. The program presented 50 trials of the red square moving across the screen. Subjects had to come into contact with the red square to receive a pellet. Touches to other locations on the screen did not yield reinforcement. The standard enrichment device involved a very large PVC tube that had holes through which food would fall when the device was shaken or manipulated. For one trial, chow biscuits were used in the standard enrichment device (this is the standard food type used with this enrichment), and the flavored pellets were used in the computer; however, subjects only interacted with the computer. Subsequent trials involved the use of flavored pellets in both devices to equate the food preferences for a total of 3 trials for each subject. Although the exact rate of delivery could not be equated, the standard enrichment would yield a large number of pellets when shaken in a particular manner, whereas the computer delivered a single pellet for each correct response. The initial choice (bear approaching and interacting by touching in any way) was recorded as well as the duration of interaction until the computer program had completed or 10 minutes had elapsed, whichever occurred first.

## 3. Results

### 3.1. Computer Training

Subjects were successfully trained to use the touchscreen computer apparatus (see Figure 1). This was accomplished using the shaping procedure described above (see Method). Subjects sometimes activated the screen with their claws, but this required pushing claws through the mesh divider, and it was much easier to use their tongues. By the end of training, subjects almost always used their tongues, but any response that broke the plane of the screen in the correct location would yield food. Subjects were also able to effectively touch a moving stimulus with their tongues despite the mesh between the bear and the screen itself. Subjects were able to interact effectively with the computer within several sessions. 

### 3.2. Behavioral Observations

This analysis focused on stereotypic behaviors (pacing, backwards walk, and abnormal behavior). Xander showed a significant increase in the amount of time spent pacing after the computer was introduced (*M* = 11.1 (out of 30 min), *SD* = 8.4) compared to before (*M* = 8.2 min, *SD* = 8.2), *t*(144) = −2.1, *p* = 0.038. The duration of pacing did not change for Sabah before (*M* = 4.1 min (out of 30 min), *SD* = 6.1) and after the introduction of the computer (*M* = 6.1, *SD* = 5.6), *t*(130) = −1.8, *p* = 0.07 (see Figure 2). Neither subject exhibited a significant difference in time spent engaged in abnormal behavior or backwards walk (in all cases, *p* > 0.05). Figure 3 depicts the average activity budget for each subject before and after the introduction of the computer, excluding low duration (<1% of time) behaviors.

### 3.3. Preference Assessment 

We first ran a trial in which the computer yielded pellets and the traditional enrichment yielded chow. There was no interaction with the traditional enrichment, so we used pellets for the remaining three comparison trials. In 100% of trials, both subjects chose to interact with the computer first, even though on some trials the bear investigated the traditional enrichment device before initiating interaction with the computer. There were also trials in which the subjects would briefly engage with the traditional enrichment device and the reengage with the computer. Overall, Xander spent 64.5% of time interacting with the computer and Sabah spent 85.7% of her time interacting with the computer (see Figure 4). 

## 4. Discussion

Overall, the outcomes of this project were highly productive and suggestive of an important avenue of future research for both positive and negative influences of computerized testing in bears. Sun bears readily responded to a touchscreen computer using their tongues. This finding highlights the importance of selecting species-appropriate forms of interaction with cognitive apparatus. In the wild, sun bears use their tongues to forage for insects and sap, and this was an ideal way to test this species. Comparative cognition research must strive to balance using standardized testing apparatus with providing opportunities for species-appropriate methods of expression. Computerized testing is an excellent way to achieve this balance. 

The training involved a standard shaping procedure in which successive approximations towards the target behavior were made over time. Initially, any touch to the computer screen yielded food reinforcement and an auditory tone. Response requirements gradually shifted towards a more specific response until subjects could eventually contact a small, moving stimulus on the screen. This type of training procedure is commonly used in zoos for training animals to participate in husbandry or veterinary procedures and will likely be well received by zoo professionals in future efforts to train animals to use computers for cognitive testing. 

Importantly, the bears’ overall response to the computer apparatus was positive. Participation was entirely voluntary, but there was never a session in which the bears chose not to participate. When given an opportunity to interact with either the computer or traditional enrichment, subjects always chose the computer first, and both subjects spent the majority of their time with the computer. Together, these results suggest that this was a positive experience for the animals. Animal welfare is an enormously complex issue for zoo management. Traditional forms of enrichment may suffer from habituation. Interest wanes quickly as novelty wears off. Computers provide a source of enrichment that is unlimited in its capacity to evolve, change, and provide more or less challenge as necessary [23,24,25,26]. It should be emphasized, however, that computers are best conceived of as complementary to other, more traditional forms of enrichment, rather than as replacements. As described, they do offer benefits relating to novelty and variation, but they also require more money and time, both of which are often limited. Thus, it is critically important to find a balance between traditional forms of enrichment and more novel and technologically advanced ones.

There was a significant increase in stereotypic pacing for one subject after the introduction of the computer. Subjects were tested on the computer in the indoor enclosure space, and one animal at a time was shifted off of exhibit. Thus, the animal remaining on exhibit could likely hear the computer program while still on exhibit, leading to the possibility that this might have inadvertently increased the duration of pacing in anticipation of using the computer. Much of the pacing behavior occurred in the vicinity of the shift door (personal observation) and may have been anticipatory in nature. This finding introduces an interesting question on how the removal of enrichment (or provisioning of enrichment to a conspecific) might influence welfare in potentially negative ways. One possible approach to addressing this question would be to compare computerized enrichment that takes place on exhibit versus inside the indoor enclosure space. Alternatively, future research might specifically focus on pacing or other stereotypic behaviors during the exact testing time of conspecifics to examine the potential stress inducing effects of another individual interacting with a preferred enrichment device. 

Overall, this research adds a new species to comparative cognition research that will yield important insights into the evolution of cognition and seems to be beneficial to animal welfare in that subjects showed a preference for interacting with the touchscreen computer. As previously discussed, there is a great deal of variability across bear species in several socioecological factors, and future research might allow important insight into the relationship between these factors and cognition. Sun bears are unique among bear species in that they follow a polyestrous, nonseasonal breeding cycle [27,28], and this might allow for specialized cognitive adaptations. Sun bears also exhibit a truly omnivorous diet consisting of a range of food items including honey, sap, insects, small mammals, and foliage. The cognitive mechanisms underlying this rich adaptability to different food sources are likely complex and varied. Future work should fully investigate the cognitive capacities of this species and continue to assess the positive and negative influences of enrichment on animal welfare.

## 5. Conclusions

Sun bears readily interacted with a touchscreen computer and learned to effectively use it within several sessions. They always chose to participate when it was available and showed some preference for it over more traditional forms of enrichment. One subject did show an increase in pacing behavior after the introduction of the computer, so future work should address anticipatory pacing as one potential issue. Overall, computerized testing appears to be a valuable avenue for future research on sun bear cognition and enrichment.

## Figures and Tables

**Figure 1 behavsci-06-00019-f001:**
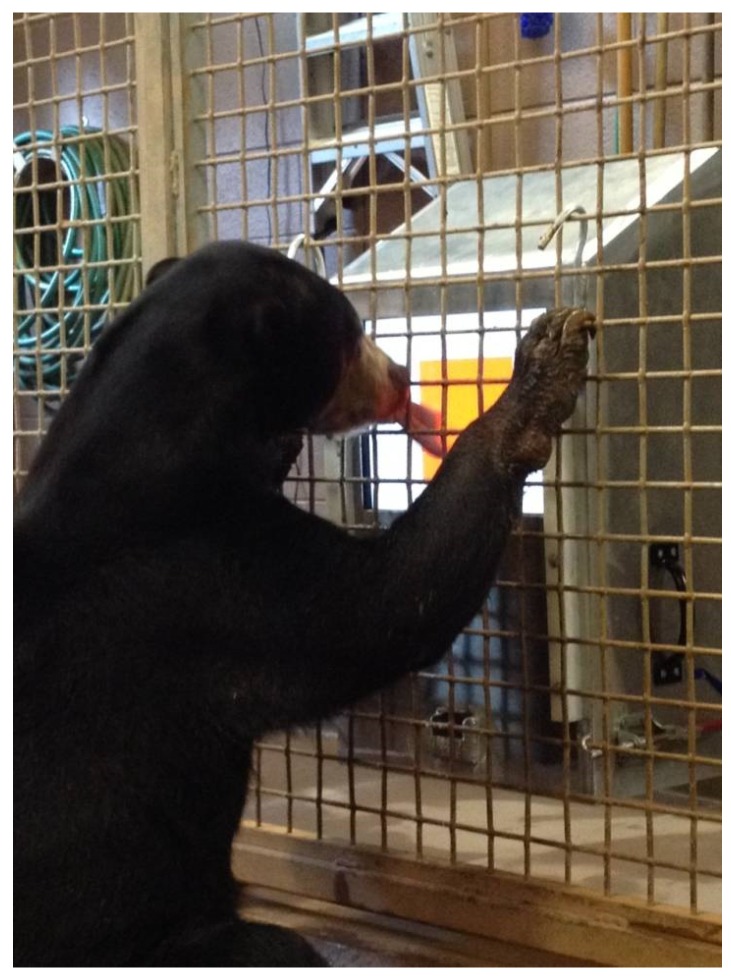
Sun bear interacting with touchscreen computer.

**Figure 2 behavsci-06-00019-f002:**
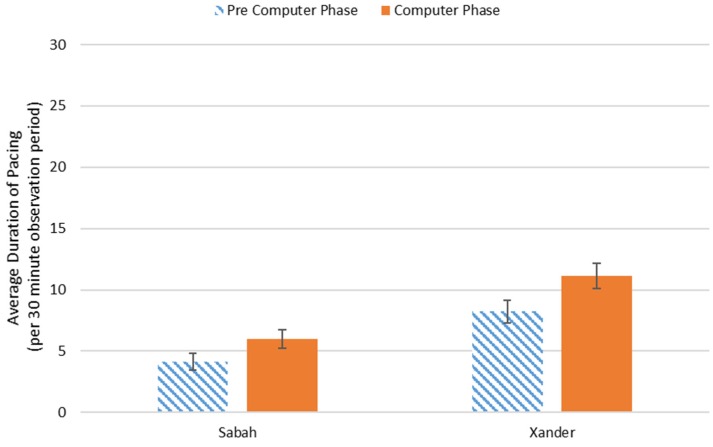
Average amount of time spent pacing before and after the introduction of the touchscreen computer (bars indicate standard error). Xander showed a significant difference across conditions at the *p* < 0.05 level.

**Figure 3 behavsci-06-00019-f003:**
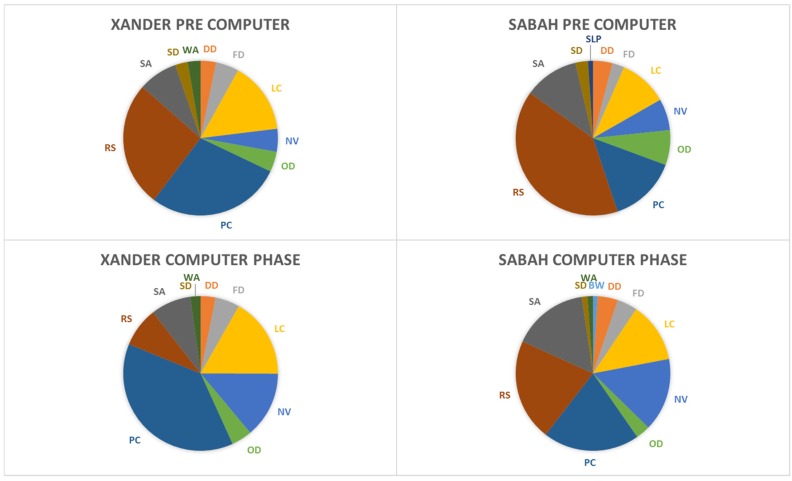
Activity budgets before and after the introduction of the computer (DD = Door Directed; FD = Feed; LC = Locomote; NV = Not Visible; OD = Object Directed; PC = Pace; RS = Rest; SA = Stationary Alert; SD = Self-Directed; SLP = Solitary Play; WA = Water Interaction). Any behaviors accounting for less than 1% of the subject’s time were not included in the figure.

**Figure 4 behavsci-06-00019-f004:**
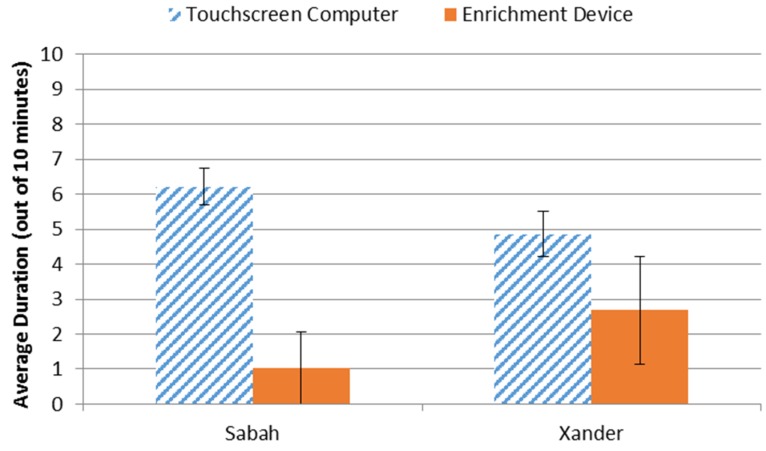
Average number of minutes interacting with either the touchscreen computer device or the traditional enrichment device.

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
