# Peer review of "The Effect of Computerized Testing on Sun Bear Behavior and Enrichment Preferences"

_behavsci, 2016, doi:10.3390/bs6040019_

Round 1

Reviewer 1 Report

Sun bears have been little studied and this paper is useful in that regard. While the use of a computer screen is not novel the evaluation of the other behaviors is of interest. It seemed as if the anticipation to use the computer affected the other behaviors. All in all this is a well-written concise paper needing only minor revision as noted below.

The problem with habituation of objects in bear enrichment was explicitly pointed out by Burghardt in the 1975 NAS zoo research volume (see recommendation number 5). That computers can more easily shift tasks and avoid boredom is an important idea and supported by this study. The preference data bare this out as well, and is a nice addition. It would be a mistake, I believe, to use computer activities as a substitute for providing opportunities for manipulation and problem solving with physical objects and this should be noted.

1) lines 33-36, While the Vonk lab reports are obviously highly relevant and important, the ignoring of the earliest experimental work on bear cognition in Ursus by Bacon and Burghardt needs to be rectified. Overviews are also available in book chapters from 1975 and 1992.

2) lines 427-430, Ethogram is a fine addition. However, to separate social play from wrestling, chasing, frolicking, or leaping on the basis of being 'disjointed' is not valid and the play literature should be consulted. In addition, none of the other four behaviors listed are themselves even in the ethogram!  This must be fixed.

3) CIs should be added to Figure 2 and 4.

4) In Figure 3 the colors should be consistent for the same behavior in all pie charts.

Author Response

Sun bears have been little studied and this paper is useful in that regard. While the use of a computer screen is not novel the evaluation of the other behaviors is of interest. It seemed as if the anticipation to use the computer affected the other behaviors. All in all this is a well-written concise paper needing only minor revision as noted below.

RESPONSE: Thank you for your positive feedback.

The problem with habituation of objects in bear enrichment was explicitly pointed out by Burghardt in the 1975 NAS zoo research volume (see recommendation number 5). That computers can more easily shift tasks and avoid boredom is an important idea and supported by this study. The preference data bare this out as well, and is a nice addition. It would be a mistake, I believe, to use computer activities as a substitute for providing opportunities for manipulation and problem solving with physical objects and this should be noted.

RESPONSE: We have added this reference. Also, this is an excellent point that computers should not be considered as a replacement for more traditional forms of enrichment. I fully agree with this sentiment and have added text to the manuscript expressing this. I have added the following statement: It should be emphasized that computers are best conceived of as complementary to other, more traditional forms of enrichment, rather than as replacements. As described, they do offer benefits relating to novelty and variation, but they also require more money and time, both of which are often limited. Thus, it is critically important to find a balance between traditional forms of enrichment and more novel and technologically advanced ones.”

1) lines 33-36, While the Vonk lab reports are obviously highly relevant and important, the ignoring of the earliest experimental work on bear cognition in Ursus by Bacon and Burghardt needs to be rectified. Overviews are also available in book chapters from 1975 and 1992.

RESPONSE: These references have been added. In the original drafting of the manuscript, the focus was largely on computerized testing which is why some of the earlier work was not included, but this has been remedied. I appreciate this important point.

2) lines 427-430, Ethogram is a fine addition. However, to separate social play from wrestling, chasing, frolicking, or leaping on the basis of being 'disjointed' is not valid and the play literature should be consulted. In addition, none of the other four behaviors listed are themselves even in the ethogram!  This must be fixed.

RESPONSE: In retrospect, I fully agree that these changes would have improved the ethogram. However, this is the ethogram that was used for the study and thus it is the appropriate information to include in the manuscript.

3) CIs should be added to Figure 2 and 4.

RESPONSE: I have added standard error bars.

4) In Figure 3 the colors should be consistent for the same behavior in all pie charts.

RESPONSE: This change has been made.

Reviewer 2 Report

This study reports the first account of training sun bears to use a touch screen computer, in prospect of applying a method to investigate the cognitive abilities of sun bears. Two adult sun bears were gradually introduced to interact with a touch screen computer, using their tongue. It was investigated if the bears prefered the computer over a standard form of enrichment and if the interaction with the computer had an effect on stereotypical behaviour. The results showed that both bears were successfully trained on interacting with the touch screen computer. The recording of initial interactions revealed that both sun bears consistently chose the computer over the other form of enrichment. They also interacted with the computer for a longer time. One of the bears showed a significant increase of pacing behaviour after the computer was introduced.

The main issue with this study is that the actual cognitive tests are missing. If the author wants to continue to discuss cognition research implications in this work, then a simple cognitive study would need to be carried out (e.g., testing the ability to differentiate between shapes or colours). There is also the possibility to discuss this work on a noncognitive level by focusing on enrichment (and welfare). At the current state, however, this study is incomplete and a major revision seems necessary.

Additional comments:

Introduction:

p. 3 (line 61): word repetition („this“)

Methods:

p.4 (line 82-85): Some more information about the subjects could be provided. Were the bears raised in the Zoo? Were they previously involved in any testing? How many other bears were they able to interact with?

p.5 (line 111): This is missing a short description of the stimuli used in the two-choice discrimination task.

 p.5 (line 128): The preference assessment is not entirely convincing. First of all it would be important to note, how often the bears were previously exposed to the standard enrichment. As mentioned in the discussion (p. 8 line 197), novelty might affect the bears preference. For a proper comparison it would also be important to know how often they interact with the standard enrichment in absence of the computer, to rule out the possibility that they generally did not interact much with this other form of enrichment.

It would further be necessary to mention what kind of bait the standard enrichment usually contained. Though it was mentioned that the standard enrichment contained the pellets in the last two trials, the fact that it was previously baited with a different treat raises the question whether the bears were just not expecting the pellets to be hidden in the standard enrichment and hence the preference of the computer was merely a food preference. To avoid this assumption, it should be mentioned in how far it was ensured that the bears were aware that the pellets were parts of both enrichment forms (e.g. were they able to smell the pellets?).

Results:

p. 6 (line 144): This is missing some descriptive information (e.g. how long did the bears need to complete the individual stages?). For establishing the use of a touch screen computer as a method in researching bear cognition such information would be highly valuable.

Figure 2: The figure is missing the indication of the significant difference in Xander's pre / computer phase. The description of the figure (line 297-298) needs to include the significance level.

Figure 3: The pie charts lack information. It does not appropriately visualize the finding that there is no significant difference before and during the computer phase. A table with the percentage of the exhibited behaviour types plus the according p values would be more informative.

p. 7 (line 171): Is the difference in time spent interacting with the computer vs the standard form of enrichment significant? The values for the other form of enrichment should either be mentioned here or included  in Figure 4.  

It would also be interesting to know if the bears ever succeeded in yielding all the pellets from the standard enrichment or if they stopped interacting with it despite the fact that it still contained pellets. This could strengthen the claim that they preferred interacting with the computer.   

Discussion:   

p.8 (line 187): This statement seems a little one sided. The discussion should also critically address the downside of a computer as an enrichment tool (costs, handling etc.) and contrast it with its benefits.

p. 8 (line 200): The discussion should address possible reasons why there were no decreases in stereotypical behaviour. It could give important indications for future research on the use of touch screen computers as an enrichment tool.  

p. 8 (line 202): Is it possible to include a proper comparison of the testing times with the times the bear was pacing? If they correlate, this could strongly support the claim. Otherwise the discussion should also address the possibility that the introduction of the computer might have induced stress in the animal.

Author Response

This study reports the first account of training sun bears to use a touch screen computer, in prospect of applying a method to investigate the cognitive abilities of sun bears. Two adult sun bears were gradually introduced to interact with a touch screen computer, using their tongue. It was investigated if the bears prefered the computer over a standard form of enrichment and if the interaction with the computer had an effect on stereotypical behaviour. The results showed that both bears were successfully trained on interacting with the touch screen computer. The recording of initial interactions revealed that both sun bears consistently chose the computer over the other form of enrichment. They also interacted with the computer for a longer time. One of the bears showed a significant increase of pacing behaviour after the computer was introduced.

The main issue with this study is that the actual cognitive tests are missing. If the author wants to continue to discuss cognition research implications in this work, then a simple cognitive study would need to be carried out (e.g., testing the ability to differentiate between shapes or colours). There is also the possibility to discuss this work on a noncognitive level by focusing on enrichment (and welfare). At the current state, however, this study is incomplete and a major revision seems necessary.

RESPONSE: Although the presented task is a very basic form of a cognitive task (locate and touch a moving stimulus), the entire process of learning to use and interacting with a computer involves a number of more advanced cognitive abilities (memory, coordination, tracking, etc). In the future, I hope to introduce more complex cognitive tasks, but at this point the simple task of tracking a target, in addition to the other components of the task, provides a useful form of cognitive stimulation for the animals.

Additional comments:

Introduction:

p. 3 (line 61): word repetition („this“)

RESPONSE: This change has been made.

Methods:

p.4 (line 82-85): Some more information about the subjects could be provided. Were the bears raised in the Zoo? Were they previously involved in any testing? How many other bears were they able to interact with?

RESPONSE: I have added this information: “These two subjects have been housed at Zoo Atlanta since 2010 in the Trader’s Alley exhibit. They are socially housed, but separated for testing. Participation in testing was voluntary and relied on subjects shifting into the test enclosure (one of their indoor enclosures). No formal cognitive tasks have been presented to these subjects, but they do engage in regular training and enrichment protocols.”

p.5 (line 111): This is missing a short description of the stimuli used in the two-choice discrimination task.

RESPONSE: I have added this information.

 p.5 (line 128): The preference assessment is not entirely convincing. First of all it would be important to note, how often the bears were previously exposed to the standard enrichment. As mentioned in the discussion (p. 8 line 197), novelty might affect the bears preference. For a proper comparison it would also be important to know how often they interact with the standard enrichment in absence of the computer, to rule out the possibility that they generally did not interact much with this other form of enrichment.

RESPONSE: The bears are regularly exposed to enrichment devices like the one used in the preference assessment, but these are rotated frequently so that one object is not regularly presented (or left in the exhibit space). At the time of the preference assessment, subjects had been testing on the computer for a year and a half, so the computer itself was not completely novel.

It would further be necessary to mention what kind of bait the standard enrichment usually contained. Though it was mentioned that the standard enrichment contained the pellets in the last two trials, the fact that it was previously baited with a different treat raises the question whether the bears were just not expecting the pellets to be hidden in the standard enrichment and hence the preference of the computer was merely a food preference. To avoid this assumption, it should be mentioned in how far it was ensured that the bears were aware that the pellets were parts of both enrichment forms (e.g. were they able to smell the pellets?).

RESPONSE: The standard enrichment is usually loaded with chow biscuits, a preferred food item for these individuals. After the first trial with the chow, I ran three trials using the testing pellets in both devices. Subjects were able to smell the pellets in the enrichment (in fact, they sometimes smelled the traditional enrichment first, then left it behind without interacting to first use the computer). I have added details to the manuscript to clarify.

 Results:

p. 6 (line 144): This is missing some descriptive information (e.g. how long did the bears need to complete the individual stages?). For establishing the use of a touch screen computer as a method in researching bear cognition such information would be highly valuable.

RESPONSE: The bears were effectively using the computer within several sessions. This has been added to the manuscript.

Figure 2: The figure is missing the indication of the significant difference in Xander's pre / computer phase. The description of the figure (line 297-298) needs to include the significance level.

RESPONSE: I have added this information to the caption.

Figure 3: The pie charts lack information. It does not appropriately visualize the finding that there is no significant difference before and during the computer phase. A table with the percentage of the exhibited behaviour types plus the according p values would be more informative.

RSEPONSE: I have modified the pie charts to better display the corresponding information using the same colors for each behavior.

p. 7 (line 171): Is the difference in time spent interacting with the computer vs the standard form of enrichment significant? The values for the other form of enrichment should either be mentioned here or included  in Figure 4. 

RESPONSE: Given the small sample size in the preference assessment, these values were not compared statistically.

It would also be interesting to know if the bears ever succeeded in yielding all the pellets from the standard enrichment or if they stopped interacting with it despite the fact that it still contained pellets. This could strengthen the claim that they preferred interacting with the computer.  

RESPONSE: Yes, in fact they sometimes interacted with it for a moment and then returned to the computer. There were also sessions in which subjects did not interact with the traditional enrichment beyond an initial investigation and the removal of a few pellets. I have added more description of this to the results section. I appreciate the recommendation of adding more information here.

Discussion:  

p.8 (line 187): This statement seems a little one sided. The discussion should also critically address the downside of a computer as an enrichment tool (costs, handling etc.) and contrast it with its benefits.

RESPONSE: I have changed the first statement to read as follows: “Overall the outcomes of this project were highly productive and suggestive of an important avenue of future research for both positive and negative influences of computerized testing in bears.

I have also added more on the downsides of computerized testing as enrichment: It should be emphasized that computers are best conceived of as complementary to other, more traditional forms of enrichment, rather than as replacements. As described, they do offer benefits relating to novelty and variation, but they also require more money and time, both of which are often limited. Thus, it is critically important to find a balance between traditional forms of enrichment and more novel and technologically advanced ones.”

p. 8 (line 200): The discussion should address possible reasons why there were no decreases in stereotypical behaviour. It could give important indications for future research on the use of touch screen computers as an enrichment tool. 

RESPONSE: Lines 218-226 describe the most likely (in my view) possibility underlying why we didn’t see a decrease in pacing behavior. I have added a suggestion that touchscreens on exhibit might alleviate this issue.

p. 8 (line 202): Is it possible to include a proper comparison of the testing times with the times the bear was pacing? If they correlate, this could strongly support the claim. Otherwise the discussion should also address the possibility that the introduction of the computer might have induced stress in the animal.

RESPONSE: I have added this text to the manuscript: “Alternatively, future research might specifically focus on pacing or other stereotypic behaviors during the exact testing time of conspecifics to examine the potential stress inducing effects of another individual interacting with a preferred enrichment device.”

Reviewer 3 Report

The author has described the use of computer touchscreens for cognitive testing and enrichment with two adult sun bears.  The protocol for shaping touchscreen use was described and a comparison between pre- and post-computer activity budgets for the bears was done.  The bears readily used the touchscreen system and appeared to prefer it to traditional enrichment devices, although an increase in anticipatory stereotypical behavior was seen in one bear as a result of introduction of the computer.

Overall, this study is easy to read and the results are clear--and clearly presented.  My questions/comments are below:

  Although the author mentions that sun bears are omnivorous in the Abstract, no mention of this (or their diet) is made in the Introduction, despite referring to the importance of this in other species.  Similarly, there is a mention of the importance of mating systems for the evolution of spatial cognition but no description of where sun bears fit into this framework.  The rationale for the importance of sun bear cognition needs to be made more claer.

 The two bears tested were 18 and 19 years old, respectively.  How long do sun bears live in captivity?

The author mentions training additional observers until 85% interobserver reliability was reached (line 120).  How many additional observers were there?  Who were they?  Why was 85% chosen?  This seems low.

Lines 144-157 of the Results section replicates the information given in the Methods section under the Computer Training heading, and so should be omitted.

The Figures (except Figure 1) don't add anything to what is already found in the text, so could be omitted.

Minor points:

unclear-- line 156 "exposed to a more 2 choice discrimination task"?

awkward wording--line 164 " was not a significant difference"

Author Response

The author has described the use of computer touchscreens for cognitive testing and enrichment with two adult sun bears.  The protocol for shaping touchscreen use was described and a comparison between pre- and post-computer activity budgets for the bears was done.  The bears readily used the touchscreen system and appeared to prefer it to traditional enrichment devices, although an increase in anticipatory stereotypical behavior was seen in one bear as a result of introduction of the computer.

Overall, this study is easy to read and the results are clear--and clearly presented.  My questions/comments are below:

  Although the author mentions that sun bears are omnivorous in the Abstract, no mention of this (or their diet) is made in the Introduction, despite referring to the importance of this in other species.  Similarly, there is a mention of the importance of mating systems for the evolution of spatial cognition but no description of where sun bears fit into this framework.  The rationale for the importance of sun bear cognition needs to be made more claer.

RESPONSE: We have added the following text to the manuscript: As previously discussed, there is a great deal of variability across bear species in several socioecological factors and future research might allow important insight into the relationship between these factors and cognition. Sun bears are unique among bear species in that they follow a polyestrous, nonseasonal breeding cycle (26-27) and this might allow for specialized cognitive adaptations. Sun bears also exhibit a truly omnivorous diet consisting of a range of food items including honey, sap, insects, small mammals and foliage. The cognitive mechanisms underlying this rich adaptability to different food sources are likely complex and varied.”

 The two bears tested were 18 and 19 years old, respectively.  How long do sun bears live in captivity?

RESPONSE: Sun bears live for up to 30 years in captivity.

The author mentions training additional observers until 85% interobserver reliability was reached (line 120).  How many additional observers were there?  Who were they?  Why was 85% chosen?  This seems low.

RESPONSE: This value was based on previous research using this vaule. The additional researchers were students trained by the Principal Investigator. Typically students trained for a month or two before reliability testing commenced. I have added more information to the method section about this.

Lines 144-157 of the Results section replicates the information given in the Methods section under the Computer Training heading, and so should be omitted.

RESPONSE: This change has been made.

The Figures (except Figure 1) don't add anything to what is already found in the text, so could be omitted.

RESPONSE: I have left them in place for now, but will remove them at the editor’s discretion pending space constraints in the volume.

Minor points:

unclear-- line 156 "exposed to a more 2 choice discrimination task"?

RESPONSE: This sentence has been removed per the recommendation above (unnecessarily duplicative of method section).

awkward wording--line 164 " was not a significant difference"

RESPONSE: I have changed this to: “Neither subject exhibited a significant difference in time spent engaged in abnormal behavior or backwards walk (all p’s > .05).”

Round 2

Reviewer 2 Report

It is important that the author shows more of a critical voice in this study. We now know that one can work with sun bears via touch screen but there is no finding in this study about bear cognition. The training can be the result of mere associative learning.

Thus study shows only potential for future research in bear cognition, it does not provide evidence on any cognitive abilities in sun bears. So, the title and the abstract seem misleading. It would improve the quality of this interesting work by making changes accordingly. Adding also one carefully placed critical statement to clarify this issue in the Introduction and Discussion is important. 

Author Response

It is important that the author shows more of a critical voice in this study. We now know that one can work with sun bears via touch screen but there is no finding in this study about bear cognition. The training can be the result of mere associative learning.

Thus study shows only potential for future research in bear cognition, it does not provide evidence on any cognitive abilities in sun bears. So, the title and the abstract seem misleading. It would improve the quality of this interesting work by making changes accordingly. Adding also one carefully placed critical statement to clarify this issue in the Introduction and Discussion is important. 

RESPONSE: Thank you for the useful feedback on the manuscript. In response to your suggestions, I have made several changes that I think will better reflect the content of the project:

The title has been changed to: 

Exploring the Role of Computerized Testing in Sun Bears

The abstract has been changed to:

The current work explored training sun bears to use a touchscreen computer which will provide one mechanism for exploring cognitive abilities in future studies. This effort also allowed for insight into a complex source of enrichment for the bears and other animals in captivity. 

I have added this to the introduction:

If sun bears can be trained to use a touchscreen computer, this would provide the opportunity for insights into the evolution of cognitive abilities as well as a potentially valuable form of enrichment for the bears. The present study is not a direct investigation of cognitive abilities, but offers an exploration of the appropriateness of computerized testing in this species. It is also an assessment of the potentially enriching influence of this technique via behavioral observations and a preference assessment. 

I have added this to the discussion:

The work presented here presents a first step in developing a method for assessing cognitive abilities in sun bears by demonstrating that they can be trained to use a computerized test station. Future work should fully investigate the cognitive capacities of this species and continue to assess the positive and negative influences of enrichment on animal welfare.

Reviewer 3 Report

I have read through this revision of the original manuscript and think that my concerns have been adequately addressed.  Focusing more on enrichment, and less on cognition, is in line with the purpose and results of the study, and makes the paper much more appropriate for publication.  I also appreciate the clarifications in the introduction and methods sections.   I am happy to recommend acceptance. 

Author Response

I have read through this revision of the original manuscript and think that my concerns have been adequately addressed.  Focusing more on enrichment, and less on cognition, is in line with the purpose and results of the study, and makes the paper much more appropriate for publication.  I also appreciate the clarifications in the introduction and methods sections.   I am happy to recommend acceptance.  

RESPONSE: Thank you for your feedback and endorsement of the manuscript!